# Making Regional Sense of Global Sustainable Development Indicators for the Arctic

**Annika E. Nilsson** [1,*] and **Joan Nymand Larsen** [2]

1    Division of History of Science, Technology and Environment, KTH Royal Institute of Technology, 114 28 Stockholm, Sweden

2    Stefansson Arctic Institute and University of Akureyri, IS-600 Akureyri, Iceland; jnl@unak.is

*    Correspondence: annika.nilsson@vetani.se

**Abstract:** Since the global Sustainable Development Goals (SDGs) were adopted in 2015, efforts are underway to identify indicators for monitoring progress. However, perceptions of sustainability are scale and place specific, and there has also been a call for Sustainable Development Goals and indicators that are more relevant for the Arctic than the global perspectives. Based on earlier and ongoing efforts to identify Arctic Social Indicators for monitoring human development, insights from scenario workshops and interviews at various locations in the Barents region and Greenland and on studies of adaptive capacity and resilience in the Arctic, we provide an exploratory assessment of the global SDGs and indicators from an Arctic perspective. We especially highlight a need for additional attention to demography, including outmigration; indigenous rights; Arctic-relevant measures of economic development; and social capital and institutions that can support adaptation and transformation in this rapidly changing region. Issues brought up by the SDG framework that need more attention in Arctic monitoring include gender, and food and energy security. We furthermore highlight a need for initiatives that can support bottom–up processes for identifying locally relevant indicators for sustainable development that could serve as a way to engage Arctic residents and other regional and local actors in shaping the future of the region and local communities, within a global sustainability context.

**Keywords:** Arctic; polar regions; sustainable development; human development; SDGs; indicators

## 1. Introduction

### 1.1. Background and Aim

Since the United Nations adopted the global Sustainable Development Goals (SDGs) in 2015 [1], major efforts have been launched to monitor progress with the help of indicators [2–4], which in turn has inspired critical assessments of indicator frameworks [5,6]. The SDGs are explicitly global in scope and meant to apply to all countries, where the indicators are the major tool for following up on the national implementation of this global policy framework. However, perceptions of sustainability are scale and place specific [7], and the situation and priorities in a region or a local community can be very different than what can be gathered from national averages. Moreover, the SDG process has been criticized for representing a top–down "cockpit-ism", followed by calls for mobilizing new agents of change rather than relying on governments and intergovernmental organizations [8]. If the goal is to engage local society and subnational decision makers in a sustainable development transition, it is thus necessary to ensure that the goals for sustainable development and the indicators to follow up on the goals are perceived as relevant for a range of local environmental and societal contexts.

While the Arctic is a region of great diversity, environmentally as well as regarding the social context in which people live, the region has been in focus for several assessments of environmental and social change at the circumpolar scale that highlight the unique features in common compared to regions further south. These relate to the cold climate but also to the fact that many people live in remote and sparsely populated areas and have maintained close ties to nature for their daily lives and livelihoods, including in settings characterized by mixed or post-industrial economy [9–11]. Arctic environments are changing rapidly due to climate change [12] at the same time as people living in the Arctic face a range of social changes related to globalization and increasing industrial interests in the region [13]. The SDGs could be a tool for navigating these changes. However, the authors of an EU-Polar White Paper about the road to the desired states of social-ecological systems in the Polar Regions argue that the global SDG framework "has not been designed with the Polar Regions in mind. As a result, UN SDGs, and their respective indicators, are not specific enough to give guidance in all decisions made concerning Polar Regions" [14]. They furthermore call for an examination of the existing SDG indicator framework in relation to what has emerged from earlier work on indicators in the Arctic. Relevant studies on social indicators for the Circumpolar North includes a project aimed at developing Arctic Social Indicators (ASIs) [15,16], the Survey of Living Conditions in the Arctic (SLICA) [13], assessments of economies of the North [17–19], studies of social-ecological resilience, including a proposal for resilience indicators [20,21], and insights about features of adaptive and transformative capacities [22,23]. The EU-PolarNet White Paper furthermore suggests that developing Arctic-relevant indicators for sustainable development requires reaching out to indigenous and local communities in the region, as well as to natural scientists working there [14].

The aim of this article is to provide an exploratory assessment of the applicability of the SDGs to the Arctic context and to discuss how the specific targets and their related indicators in the UN reporting system relate to insights from relevant Arctic assessments of human development, with emphasis on the Arctic Social Indicators, resilience, and adaptive capacity. Our assessment will also include attention to insights from scenario workshops and interviews that we have been conducting in various locations in the Barents region and Greenland from 2015 to 2019 around questions concerning the drivers of change and issues that are likely to be locally and regionally important in the coming two to three decades [24] and further as yet unpublished work, as well as questions related to what people view as important aspects of their overall sense of well-being.

The notion of Arctic SDG indicators could suggest that one set of indicators is relevant for the whole region. This is very likely not the case, given the diversity of social contexts across the Circumpolar North, ranging from small remote settlements to modern cities. Moreover, while more relevant than the global SDGs, such a set could still be seen as a form of "cock-pitism" that may not serve local and subnational decision making any better than the global SDGs. Nevertheless, we believe that a scrutiny of the global SDGs in relation to existing literature on Arctic change is relevant for two reasons. The first is that efforts are already underway to apply the SDG framework in national and subnational monitoring and policy efforts. A scrutiny of the framework is thus needed, as suggested in the EU-PolarNet White Paper. The other is the need for capacity building to support bottom–up local and subnational processes for developing locally relevant indicators for sustainable development (which may or may not be inspired by the SDG framework or the work on Arctic SDGs). We return to this issue in the discussion.

*1.2. Theoretical Context*

In the Arctic, interest in the notion of sustainable development emerged in the 1980s in parallel with and partly in response to the global discourses on how to reconcile economic and social development with environmental protection [25]. As global change research developed, scientific practices also came under scrutiny, leading to calls for "sustainability science" with more interdisciplinary approaches and an emphasis on addressing societal problems, e.g., [26]. Its further development included attention to system dynamics in resilience thinking, with focus on the capacity of social-ecological systems

to cope with and recover from disturbances and more recently developed with added attention to people's capacity to learn in order to adapt to change or deliberately try to transform the system [27–30]. Partly linked to the focus on learning as a key social aspect of social-ecological systems, sustainability science has come to include a stronger focus on processes for arriving at new development pathways, including an emphasis on co-production of knowledge [25,31]. Arctic research features a growing body of work based on local case studies where co-production of knowledge with communities has been central. While the exploratory assessment of the global SDGs featured in this article is a research-driven effort, our analysis builds on empirical work that is guided by norms from sustainability science, including engagement with various stakeholders to co-produce knowledge. As we return to this in the Discussion, we foresee a need for further strengthening of such local engagement. However, given that localities across the world are connected to global processes that both affect and are affected by local activities, there is also a need to create participatory processes in which the global-local links can be explicitly explored [24]. The global SDG framework could serve as one starting point for such processes, but only if it is perceived as relevant or can be made relevant for learning processes at the subnational and local levels.

The legitimacy of the SDG framework cannot be taken for granted. One concern is that various actors have used the plasticity of the concept of sustainable development to forward their own specific perspective, not seldom with national interests in mind, as shown in studies of Arctic sustainability discourses [32]. A second concern relates to the fact that the function of indicators reaches far beyond informing policy and they become tools of governability and managerialism [33]. Lehtonen et al. argue that instead of focusing on the instrumental functions of indicators (as guides for better decision making), more attention should be placed on "the political framework conditions shaping and being shaped by indicators" [33]. While recognizing the inherent limitations of indicators and the criticism of indicators as tools of managerialism, we consider it timely to scrutinize the SDG framework from an Arctic perspective, given that the global SDG framework is likely to figure in assessing development with potential implications of various policy decisions. However, it is an important argument for an increasing focus on process and co-production of knowledge in the further work on Arctic SDGs.

## 2. Materials and Methods

The basis for our assessment of the applicability of the global SDGs for the Arctic is the Global indicator framework for the Sustainable Development Goals and targets of the 2030 Agenda for Sustainable Development [34]. It includes 232 unique indicators. The total number of indicators in the SDG framework is 244 but nine of them are listed for more than one target. These indicators have been developed by the UN Inter-Agency and Expert Group on SDG Indicators and agreed by the UN General Assembly. They thus represent a set of politically negotiated indicators.

The Arctic-relevant features of sustainable development have been derived from several sources, as described below.

### 2.1. Arctic Social Indicators (ASIs)

The Arctic Social Indicator project is the most ambitious effort to identify indicators for human development that are relevant for the Arctic. The project followed on from the conclusion in the 2004 Arctic Human Development Report that assessment of human development in the Arctic needs attention to more factors than those that were highlighted in the Human Development Index and that a framework was needed for monitoring changes in human development in the Arctic. In addition to attention to longevity, education, and material success in the Human Development Index, the Arctic Human Development Report emphasized controlling one's own destiny, maintaining cultural identity, and living close to nature [35]. In the ASI project, these ideas were further developed in an engagement process that included workshops with indigenous representatives and other Arctic stakeholders as well as close collaboration with representatives of the Arctic Council's Sustainable Development Working Group. Based on these discussions, ASI came to focus on six key domains of human development:

health/population, material well-being, closeness to nature, education, cultural well-being, and fate control. Furthermore, based on strict selection criteria on data availability, data affordability, ease of measurement, robustness, scalability and inclusiveness, a small set of seven indicators were identified as especially relevant [15], and they are:

1.  Infant mortality (Domain: health/population)
2.  Net migration (Domains: health/population and material well-being)
3.  Consumption/harvest of local foods (Domains: closeness to nature and material well-being)
4.  Per capita household income (Domain: material well-being)
5.  Ratio of students successfully completing post-secondary education (Domain: education)
6.  Language retention (Domain: cultural well-being)
7.  The fate control index (Domain: fate control)

The focus on developing a *small* suite of indicators—preferably one indicator per domain—and to choose indicators that could be readily measured using existing data placed considerable constraints on the exercise. Other indicators were also developed and assessed in the process but presented as alternatives that did not all meet the strict selection criteria.

In a second phase of the project (ASI 2010–2014), the applicability of the suggested indicators was further scrutinized and evaluated based on five regional case studies [16]. While the exercise showed the value of the approach in that it was possible to draw important conclusions about human well-being for each of five case study regions, it also made apparent the need to adjust the original indicators to the specifics of each location. It furthermore revealed that a lack of comparable data and common data protocols for comparative studies are major challenges for assessing human development across the Arctic. Furthermore, the lack of primary data, in particular data on the contribution made by the subsistence component of the northern economy, presents major challenges in obtaining more accurate and unbiased estimates of material well-being and closeness to nature. So far, the ASI framework has only focused on the social indicators, and therefore does not include, e.g., indicators related to ecosystem processes or technological features. However, revisions and extensions of the current framework are on-going with a view to consider additions, e.g., on bio-physical dimensions.

*2.2. Arctic Resilience Report*

The notion of social-ecological resilience has received increasing attention in the Arctic, where an assessment process under the auspices of the Arctic Council has contributed to mainstreaming the concept in a policy setting [29,30]. The process has included identifying sources of adaptive and transformative capacity based on published literature and categorizing these into seven essential "capitals": natural capital, social capital, human capital, infrastructure, financial capital, knowledge assets and cultural capital [22,23]. While these have not been associated with specific indicators, they can serve as reminders in identifying features that need to be included in monitoring progress towards sustainable development in the Arctic, similar to the ASI "domains".

The Arctic Resilience Report also includes an assessment of factors that build and erode resilience in the Arctic, applying a standardized template for a meta-analysis of 25 local case studies gathered from published research from a range of projects [20]. The authors highlight the "ability of societies to self-organize", followed by "nurturing diversity" and "learning to live with change and uncertainty" as the features that stood out the most. Similar to the "capitals", these features serve as a reminder of aspects that need to be in focus when defining sustainable development and assessing progress towards specific goals in a region that is currently experiencing rapid social and environmental changes.

In an effort to operationalize the insights from the resilience assessment, Carson et al. have tried to identify indicators that capture the interaction between the social and ecological systems. Their analysis is based on previous empirical research and applied to five case studies in the Barents area that were previously published or developed by local experts. Their analytical framework features attention to "livelihoods", "knowledge/learning" and "self-organization", with "diversity" and "embracing

change" as cross-cutting themes [21]. While their approach focused on inspiring local, bottom–up qualitative assessment processes, the domains that have been identified are potentially relevant as an additional check point in identifying and prioritizing among potential sustainable development indicators for the Arctic and was therefore included in our study.

### 2.3. Scenario Workshops and Interviews

As pointed out by the EU-PolarNet White paper, there is a need for local input and ownership of efforts to monitor progress towards the SDGs [14]. This is also in line with Arctic sustainability research, where quality participation by local communities and co-production of knowledge are increasingly emphasized [22]. Participatory approaches are also increasingly used in developing scenarios of potential future change in the Arctic [36]. Our own work in this field includes a combined bottom–up identification of locally relevant drivers of change with discussion about how these might play out in different potential global futures based on the Shared-Socioeconomic Pathways (SSPs) that are used for the IPCCs scenarios [23]. The scenario exercises have been guided by the question: What future changes may influence this region economically, environmentally and socially within the perspective of one to two generations? The scenario material is relevant in that it highlights the need for attention to new issues that might emerge in the near future and should be captured in a set of indicators for SDGs for the Arctic. In our mapping of issues that need to be addressed in SDG indicators for the Arctic, we draw on previously published results from an initial set of four scenario exercises in 2015 (in Pajala, Sweden; Bodø, Norway; Kirovsk, Russia; and with young reindeer herders across the Eurasian North [24,37]) and on as of yet unpublished observations from participatory scenario work (in Ilulissat, Greenland; Kiruna, Sweden; and Alta, Norway) and focus groups, and individual semi-structured interviews in Greenland (Ilulissat, Arsuk, Narsaq) in 2017–2019. In our analyses, the data from the workshops were compiled in accordance with elements of the global SSPs [38], while maintaining an eye towards issues that were deemed important by workshop participants but were poorly captured in the SSP framework.

### 2.4. Assessment Process

The assessment of the SDGs in relation to their relevance for the Arctic was carried out by examining each of the global SDGs (including targets and indicators) in relation to insights from the processes described above, looking for matches both at the level of goals and target in the SDGs and domains or features identified as relevant for the Arctic. For each of the SDG targets, a judgement was made about whether it represented some or significant overlap with the suggested small suite of ASIs and if major issues highlighted in the ASI or in the other processes were missing. When relevant, we have added attention to issues discussed in the ASI report but not selected as part of the small suite of indicators. In the analyses we also draw on our own insights from working with developing the ASIs (Larsen) and in the Arctic Resilience Report (Nilsson).

## 3. Results

In this section, we summarize the insights from the Arctic-specific assessments of each of the 17 global SDGs, including attention to targets and indicators.

### 3.1. SDG 1. End Poverty in All Its Forms Everywhere

While the top-level text for Goal 1 is about poverty, the related targets also highlight social protection measures (SDG Target 1.3), equal access to resources (SDG Target 1.4) and building resilience to climate-related extreme event and other economic, social and environmental chocks (SDG Target 1.5). The income-related targets match well with the ASI domain "material well-being". While the SDG indicators focus specifically on percentage of population below a certain income level, the closest ASI indicator is per-capita income, from which information for the SDG indicators could be derived. In the initial ASI report, Section 2.2.4 discusses poverty as a possible ASI material well-being indicator [15].

However, in many Arctic contexts, people rely heavily on subsistence/non-market activities for their daily needs [39]. As argued in the ASI, focusing exclusively on monetary income is therefore insufficient, and without primary data collection on subsistence and traditional economy, it would be difficult to obtain a reliable measure of poverty. The ASI indicator "consumption/harvest of local foods" would help capture the importance of subsistence activities but requires primary survey data, similar to the work carried out by the Survey of Living Condition in the Arctic [40]. Closely related to consumption of local foods is the right to land and the right to harvest living resources, which would also need to be captured by a set of Arctic-relevant SDG indicators. These relate to the ASI domains "fate control" and "closeness to nature". For fate control, the ASI team has proposed a composite index that includes attention to political power, decision-making power, economic control and knowledge construction control [15], which is more encompassing than the attention in the SDG framework to land tenure being legally recognized and secure. It could, for example, include attention to meaningful participation in co-management of natural resources and in the co-production of knowledge relevant for managing the resources that form the base of the subsistence economy.

SDG 1 and its related indicators have a strong individual/household focus. Insights from scenario workshops in Sweden also highlight a need to look at municipal income, or tax base. Municipalities often have some responsibility for basic services and it is not uncommon that municipalities in the Arctic have a weak local tax base, because of few people and low incomes due to a demography shaped by outmigration of the working age cohorts, combined with high costs because of distances and low population densities.

The focus on building resilience and on disaster risk reduction (SDG Target 1.5) is very relevant for many parts of the Arctic, where climate-related changes already affect living conditions, such as erosion forcing relocation of settlements, and livelihoods, such as extreme weather with severe consequences for reindeer herding [41]. The most relevant SDG indicator would be "proportion of local governments that adopt and implement local disaster risk reduction strategies in line with national disaster risk reduction strategies" (SDG Indicator 1.5.4). It would align well with the increasing attention to adaptation actions in a changing Arctic [42–44].

## 3.2. SDG 2. End Hunger, Achieve Food Security and Improved Nutrition and Promote Sustainable Agriculture

In a region where transport infrastructure is underdeveloped and vulnerable to weather, and where foods in stores can be very expensive, food security is often related to access to country foods. Food security is increasingly being challenged by climate change, globalization, and industrial development, which is reducing both access to and availability of country food [41]. It is an issue that has also been highlighted in several scenario workshops as important for a viable local future. The ASI indicator "consumption/harvest of local foods" is thus an important complement to SDGs. It should also be noted that Arctic indigenous peoples are developing their own definitions of food security that are much more holistic than measures that relate to nutrition from a medical point of view, e.g., [45]. In further discussions of relevant indicators, it would thus be pertinent to engage with indigenous peoples in various parts of the Arctic to identify what features of society and nature would be most appropriate to include in assessment of SDG 2. Furthermore, more work on food security, sustainable agriculture, bio economy and blue economy, and economic sustainability in the Arctic is needed [46]. The gap in knowledge on food security was also highlighted in the 2014 follow-up to the original Arctic Human Development Report [13].

SDG 2 includes indicators that relate to sustainable farming. While farming is relevant in some parts of the Arctic and may indeed become more important with a warmer climate, other aspects of the food production system also need attention. These include fishing, hunting, and herding systems—all of which are affected by climate change, in addition to direct interactions with human activities. In some parts of the Arctic, interaction with other activities include growing pressure from industries, such as extraction of non-renewable resources, leading to increasing competition for land as well as risks related to pollution with potentially long-term consequences for local food security [47].

Fisheries are covered in SDG 14. "Conserve and sustainably use the oceans, seas and marine resources for sustainable development", while herding and hunting systems are poorly covered in the SDG framework. To monitor the environmental base for food security in the Arctic, it would be relevant to explore an indicator that highlights changes in land, lakes and rivers as well as coastal and marine areas that are available for food production whether it be fisheries, herding, hunting, or agricultural systems.

### 3.3. SDG 3. Ensure Healthy Lives and Promote Well-Being for All at All Ages

SDG 3 and its related indicators capture a range of specific health issues, including infectious diseases, mental health, substance abuse, deaths due to accidents, and exposure to pollution—all of which are also relevant in the Arctic. Some of the same issues are covered in the ASI domain "health and population" where the ASI framework has identified infant mortality as an indicator that "relates directly to quality of life and people's sense of well-being, and it integrates a wide range of health-relevant conditions including health infrastructure, sanitation, nutrition, behavior, social problems and nutrition, behavior, social problems and disease" [16]. The ASI work also list alternative indicators, similar to those highlighted by the SDG. Our interview results show that mental health is increasingly becoming a more relevant social indicator of the Arctic. It should be noted that human health in the Arctic is already assessed by the Arctic Monitoring and Assessment Programme (with focus on pollution issues, climate change and nutrition) [48] and in the Arctic Human Development Reports (with a broad focus on health and well-being) [49]. The experts responsible for these assessments would be well positioned to help identify health indicators that should be prioritized as part of an Arctic SDG framework.

SDG 3 includes attention to health care infrastructure, which is a challenge in many parts of the Arctic due to low population densities and large distances. One suggested SDG indicator is "health worker density and distribution" (3c1). Our empirical data also highlight that getting to health care providers can be both expensive and time consuming for residents in the Arctic. Moreover, elderly care for an aging population can be a challenge for municipalities due to lack of personnel. The latter is partly connected to a demographic situation in some parts of the Arctic where old people stay in rural areas and small towns while young people move south or to urban areas. It highlights a lack of attention to demography in the SDG framework, which in the ASI framework is captured by an indicator focusing on outmigration.

### 3.4. SDG 4. Ensure Inclusive and Equitable Quality Education and Promote Lifelong Learning Opportunities for All

The importance of education and learning capacities are recognized in all material that we have included as background for the analysis. In the ASI framework, it is captured by focusing on the ratio of students successfully completing post-secondary education opportunities. The argument is that "[p]articipation in and completion of post-secondary education opportunities is one sign of a healthy community, and as such can serve as a reliable indicator of the general role of education in terms of contributing to the well-being of Arctic communities" [16].

While formal education is an important aspect of the knowledge assets and human capital needed to support adaptive and transformative capacity, other forms of knowledge are also important in the Arctic, not least indigenous knowledge. The 2014 Arctic Human Development Report highlights among key trends and Arctic success stories the increasing use of indigenous knowledge in formal education and the growing recognition of local and indigenous knowledge in many parts of life in the Arctic [13]. Furthermore, the role of indigenous knowledge is increasingly recognized in both international conventions (e.g., Article 8 (j) in the Biodiversity Convention), in assessments carried out under the auspices of the Arctic Council, and as an aspect of indigenous peoples' rights. It has also become part of the discussion about education in the Arctic, challenging earlier divisions between formal and informal education [50]. While the role of indigenous knowledge is difficult to capture by an indicator, the ASI's attention to "language retention" as an indicator of cultural well-being is

a potential candidate. Language retention here refers to the number or percentage of speakers of ancestral language.

A major challenge for education in the Arctic is access to educational opportunities within a reasonable distance from home, where rural and remote communities can find themselves in a vicious circle where lack of schooling opportunities lead to out-migration, leaving too few students to keep the school open [50]. While the global SDG framework includes indicators that highlight access to schools with basic material facilities and teachers, they address neither the concerns of culturally relevant education nor the issue of having to leave one's home community to attend school. Such indicators could be important for guiding decisions that would affect achievements in formal schooling as well as human capital and knowledge assets that include attention to indigenous and local knowledge.

### 3.5. SDG 5. Achieve Gender Equality and Empower All Women and Girls

The ASI reports discuss gender as an important aspect of several of the selected indicators but does not include any indicator that specifically matches SDG 5. Furthermore, the concluding chapter of the 2014 Arctic Human Development Report observes that there are still significant gaps in knowledge about gender in the Arctic, including insufficient attention to gendered aspects of impacts of Arctic change and of geopolitics [13,51]. It is well known that demographic patterns and out-migration have gendered dimensions, where women are more likely to leave and where resource industries have historically attracted male workers [52,53]. The SDG indicators specifically highlighting gender equality focus on issues related to discrimination, representation in decision-making, individual fate control and violence, which are issues that were also highlighted in the chapter Gender Issues in the first Arctic Human Development Report [54]. Unlike the SDG framework, this chapter also highlight a need to analyze men's changing role in society and how it affects violence towards self and others. Further discussion is thus needed on Arctic SDG indicators that would capture gender dimensions of society as they relate to both women and men in the Arctic.

### 3.6. SDG 6. Ensure Availability and Sustainable Management of Water and Sanitation for All

Drinking water and sanitation are highlighted as emerging issues in the 2014 Arctic Human Development Report [13] but are not part of the ASI framework. They could be viewed as part of overall "infrastructure" and "natural capital" as important features of adaptive and transformative capacity. It is a type of infrastructure that is vulnerable to both climate change (e.g., permafrost thaw, changes in precipitation patterns, introduction of new pathogens) and where sources of drinking water today and in the future need protection from pollution from industrial activities. Furthermore, a recent survey reported that "many remote Arctic and sub-Arctic residents lack WASH services, and these disparities are often not reflected in national summary data" [55]. Such lack of services was reported from respondents in all Arctic states. Access to safe drinking water was also raised in relation to pollution and climate change in a scenario workshops in Kirovsk, Russia. It thus appears pertinent to further discuss Arctic-relevant indicators that would capture developments in relation to SDG 6, including how achieving this goal might be affected by future environmental changes and industrial developments.

### 3.7. SDG 7. Ensure Access to Affordable, Reliable, Sustainable and Modern Energy for All

Access to affordable and sustainable sources of energy varies greatly across the Arctic, where most people in the Fennoscandian North and Iceland are connected to grids that rely on a relatively large proportion of renewable energy sources. By contrast, remote Arctic communities in Canada, Alaska, Greenland and Russia often rely on fossil fuels affected by "high transportation and commodity prices, lack of transportation infrastructure, high environmental and human health risks, and other costs." [56,57]. Energy has thus become an important issue not only related to climate change mitigation but also to high costs of living in the North. Many remote communities still rely on outdated diesel generators for their electricity supply, with associated high fuel costs and local pollution (in addition to

the contribution to greenhouse gas emissions). There is a growing interest in alternative energy sources that are suitable for off-grid operation, such as solar and wind [58]. However, there are also limitations, such as financial subsidies of fossil fuels for energy generation, large costs of renewable projects, and insufficient financial incentives [56,57]. The ASI framework does not include attention to energy access (although plans for future updating and adjustments to the ASI indicator set will include attention to this) and it was not specifically highlighted in our own empirical material. However, several of the global SDG indicators related to affordable and reliable energy are relevant for the Arctic. Given the low number of people living in these communities compared to towns and cities, the gathering of data must be appropriately scaled to be meaningful, i.e., focusing on each community or subnational region rather than national averages.

*3.8. SDG 8. Promote Sustained, Inclusive and Sustainable Economic Growth, Full and Productive Employment and Decent Work for All*

The overall goal of SDG 8 aligns well with the ASI domain "Material well-being" and with the emphasis of "financial capital" as an important base for adaptive and transformative capacity. It also mirrors the mention in scenario workshops and interviews of employment opportunities as a major issue for development that is locally sustainable. While economic growth in the SDG framework is mainly captured by indicators related to GDP, the proposed ASI indicator for economic well-being is per capita household income, which is motivated by it providing a more accurate estimate of income in the North. The main limitation of GDP (or in the case of Arctic regions the Gross Regional Product, GRP) for measuring material well-being in the Arctic is the significant economic leakage in the form of payment to factors of production from outside the Arctic which is linked to a significant share of ownership and control being in the hands of non-regional interests, which leads GRP per capita to be overestimated [59,60]. Features from the SDG indicators that one could argue are encompassed by the per capita household income indicators are unemployment rate (8.5.2) and average hourly earnings (8.5.1). The ASI authors highlights that household income as an income indicator still ignores both direct services purchased with public transfers and production in the traditional economy [16]. These limitations of course also apply to SDG 8.5.1 and 8.5.2.

In addition to economic growth, SDG 8 highlights issues related to working conditions and worker's rights that are not captured in the ASI framework. Attention is also directed towards efficiency of the economic growth, with indicators focusing on material consumption and footprints, as well as the strength of the institutional financial framework (e.g., access to banks).

SDG 8 furthermore refers specifically to sustainable tourism (8.9); also mentioned under SDG 12b. Some parts of the Arctic are currently experiencing a major increase in tourism such as in the case of Iceland, with both social and environmental impacts, raising questions about the sustainability of this industry [61,62]. In scenario workshops, tourism has been mentioned both as an opportunity for new jobs and as a challenge to local livelihoods and the capacity of existing infrastructure. It would thus be highly relevant to include an indicator that would follow the development of this industry in an Arctic SDG framework.

With the increased emphasis on economic sustainability in the North—and as also reflected in results from scenario workshops and interviews—there is a need for more efforts to develop regional genuine progress indicators as an alternative to GRP and per capita household income. Northern economies have been assessed several times, most recently in 2015, with attention to the macro level as well as to specific features and sectors [17]. These assessments are carried out by a team of experts and researchers from national statistical offices and academic institutions across the Arctic. It would be pertinent to rely on their expertise in identifying and defining the indicators that would be most relevant for assessing how well the Arctic and different regions within it perform in relation to SDG 8. Based on comments in workshops and interviews, we also highlight a need for special attention to the boom and bust nature of many resources-based economies, who has control over the monetary

gains generated in the region, conditions for entrepreneurship, and the gendered dimensions of many local economies.

*3.9. SDG 9. Build Resilient Infrastructure, Promote Inclusive and Sustainable Industrialization and Foster Innovation*

SDG 9 combines attention to the sustainability of industrial development in relation to climate, including emission of carbon dioxide, with a range of other issues that affect economic development, such as communication infrastructure (physical and virtual), research expenditure, and proportion of small-scale, manufacturing, and high-tech industries. Taken together, the suggested SDG 9 indicators appear to provide a profile of the innovation capacity of the economy. There is no equivalent ASI indicator. In relation to factors that promote adaptive and transformative capacity, there is a match with the attention to "infrastructure", which is lacking or poor in many parts of the Arctic and often also very expensive to build. The 2015 Economy of the North report discusses the same theme as SDG 9 as "Arctic's Emerging 'Other' Economies: Technology, Knowledge and Culture in the New Arctic Economy" [17]. It concludes with a comment that with "continuing globalization, urbanization and growth of post-industrial sectors in the Arctic, these 'other economies' will be playing even more substantial role in the future" and that they are predominantly urban where they result "from the application of local human capital and other factors of production." Moreover, the analysis shows that "some of these industries have higher productivity and lesser volatility than the resource sector, and therefore are more compatible with the notion of sustainable economic development in Arctic regions." Their development is thus highly relevant to monitor in assessing how well the SDGs can be achieved in the Arctic. The 2014 Arctic Human Development Report similarly notes the increase in non-resource extractive industries, and the increase in the marketability of northern culture [63]. It should be noted that the conditions for creative economies in the Arctic are very place specific and related to existing economic structures, and it will therefore be important to keep scale in mind when gathering data to follow up on indicators [64]. Petrov creates an index that captures aspects of human capital that are especially relevant for the development of creative innovation economies, which could be considered for inclusion in an Arctic SDG framework [65].

*3.10. SDG 10. Reduce Inequality Within and Among Countries*

SDG 10 and its sub-goals address economic inequalities, regulation of financial flows and policies related to discrimination. The most relevant ASI is per capita household income [59]. A focus on inequality or poverty would—as also noted in the work of ASI—require a measure of income distribution in addition to the per capita household income. Gini coefficients would be useful measures, but they might be impractical due to the small size of communities and existing data challenges.

To meet the objectives of SDG 10, an Arctic SDG framework would also need attention to the policies that regulate the distribution of income within countries (e.g., policies that support regional economic development in northern areas). Of special relevance would be attention to the financial flows related to extractive industries, in order to highlight how much of the income stays in the region and in the local communities where extraction takes place. As highlighted by Huskey et al. "natural resource production often separates local income from production" [19].

Inequality is also about lack of political representation and discrimination—neither of which are explicit parts of the ASI framework, although political representation is included as one of the components in the ASI fate control index. Given a history of lack of representation of indigenous voices in political decision making and discrimination, along with the increasing attention to indigenous peoples' rights as an important aspect of human right, it would be relevant to monitor progress here in a systematic manner. Bankes and Koivurova discuss the ASI indicators related to cultural well-being and language retention and to fate control and as potential candidates, and specifically mention recognition of human rights [66]. Other relevant measures would include more specific recognition of indigenous rights and formal representation in relevant decision-making processes.

### 3.11. SDG 11. Make Cities and Human Settlements Inclusive, Safe, Resilient and Sustainable

This goal addresses the local context in which people live, including its safety and its provision of transport and inclusiveness. It also highlights attention to natural and cultural heritage, as well as to green spaces. Related ASI indicators are "contact with nature" and "cultural well-being". Related to features of adaptive capacity, it speaks to the need for natural capital, cultural capital and infrastructure. An issue that would de especially relevant to highlight in Arctic SDGs would be the vulnerability of cities and settlements to the impacts of climate change, including degrading permafrost, rising sea levels and associated storm surges, and extreme weather that leads to landslides and avalanches. While such concerns are regularly highlighted in assessment of the potential impacts of climate change in the Arctic, and adaptation actions are increasingly proactive, e.g., in spatial planning, barriers to adaptation are still a concern [67]. Monitoring the implementation of integrated policies and plans would be relevant, as suggested by SDG 11b, with reference to holistic disaster risk reduction management at all level. Another issue needing special attention for Arctic SDGS is the role of urbanization, which affects physical living conditions but also many other trends, such as demography, as well as the relationship between urban center, larger communities and smaller settlements [68].

### 3.12. SDG 12. Ensure Sustainable Consumption and Production Patterns

This goal is mainly about reducing waste. While it has no equivalent ASI, the potential impacts of poor waste handling in the Arctic have been highlighted in Arctic pollution assessments, e.g., emission from uncontrolled burning of waste [69] and in local scenario workshops, e.g., concern for drinking water quality [70]. Waste is also a major concern in relation to mining, where the handling of waste rock and metals processing has had major environmental impacts on local water and air quality [71], and in some cases left toxic legacies that will remain for millennia, such as at Giant Mine in Canada [72]. Past, current and potential local impacts from oil and gas production are also well documented [73]. Given the combination of extensive past activities and current interest in further mining and hydrocarbon development in the Arctic, it would be highly relevant to include an indicator that highlights the amount and type of waste generated by industry (SDG indicator 12.4.2), recycling rates (SDG indicator 12.4.2) that affects the need for new production, and sustainability practices of the production companies (SDG indicator 12.6.1). Give the relatively sparse population, the material consumption in the Arctic may not be as relevant from a global perspective but is nevertheless important for the local environment, especially in communities with poor trash handling facilities.

### 3.13. SDG 13. Take Urgent Action to Combat Climate Change and Its Impacts

Goal 13 covers adaptation to climate change, policies related to mitigation, and awareness-raising measures. While the Arctic has been called a bell-weather for global climate change based on a wealth of research and observations [12,42–44,74], the political commitment to action nevertheless varies across the Circumpolar North. The need for monitoring along the lines suggested by the SDG framework is thus highly relevant also for an Arctic SDG framework, including attention to goals 13.1 (strengthen resilience and adaptation), 13.2 (integrate climate change measures in policies, strategies and planning) and 13.3 (improve education, awareness and institutional capacity). The ASI process is working on incorporating bio-physical dimensions, including climate factors, and to contribute to the development of strategies for strengthening adaptation to global change.

### 3.14. SDG 14. Conserve and Sustainably Use the Oceans, Seas and Marine Resources for Sustainable Development

Much of the Arctic is ocean that is covered with ice either seasonally or year round and its environment is highly impacted by a warming climate, with potential impacts on marine resources. Resources from Arctic marine areas play a major role for food provision not only locally but also for national economies and globally. SDG 14 is highly relevant for an Arctic SDG framework. In relation to

existing processes, ASI's focus on "Contact with nature" and "Consumption and harvest of local foods" are relevant. Equally relevant are SDG indicators that more directly focus on the marine environment, including attention to acidification (14.3.1) where the Arctic has been highlighted as an especially sensitive region [75] and plastics (14.1.1), state of costal management plans in terms of ecosystem-based approaches (14.2.1) and protected areas (14.5.1), and indicators focusing on the status of fish stocks (14.4.1). These could be assessed and developed with more specific attention to Arctic conditions based on insights from the Arctic Marine Biodiversity Report [76].

*3.15. SDG 15. Protect, Restore and Promote Sustainable Use of Terrestrial Ecosystems, Sustainably Manage Forests, Combat Desertification, and Halt and Reverse Land Degradation and Halt Biodiversity Loss*

SDG 15 addresses the state of terrestrial ecosystems and biodiversity, where the suggested indicators focus on protected areas and species and on sustainable forestry management, and equitable sharing of benefits related to genetic resources. Similar to the situation in relation to SDG 14, the ASI framework targets human relationship to the environment rather than the terrestrial environment as such. Some of the gap would be covered by work carried out under the auspices of the Arctic Council Working Group on Conservation of Arctic Flora and Fauna (CAFF), including the recommendation from the Arctic Biodiversity Assessment to increase and focus inventory, long-term monitoring and research efforts [77,78]. Compared to the SDGs, such monitoring would naturally include attention to environments that are uniquely Arctic and how they are changing due to a warmer climate (e.g., tundra, high Arctic landscapes). The policy recommendation from the Arctic Biodiversity Assessment also highlight the need to monitor cumulative effects of stressors and drivers of change [78]. Freshwater environment, including rivers, lakes and wetlands, are not specifically covered in the SDGs but play an important role in the Arctic. Here a new assessment of biodiversity in freshwater environment may serve as a relevant starting point for complementing the global SDG indicators in an Arctic SDG framework [79].

*3.16. SDG 16. Promote Peaceful and Inclusive Societies for Sustainable Development, Provide Access to Justice for All and Build Effective, Accountable and Inclusive Institutions at All Levels*

SDG 16 is about the rule of law, reducing violence of all kinds, and effective institutions. It lacks any specific equivalent in the ASI framework but relates in part to fate control in the ASI (e.g., indicators of political power and political activism, and indicators of human rights), and on the emphasis on well-functioning institutions and on social capital in the literature on adaptive capacity. There is no reason to believe that these goals would not be equally relevant in the Arctic as elsewhere in the world, but these issues would need to be detailed further in bottom–up processes with Arctic stakeholders, including attention to access and support in rural areas far from services. Furthermore, an Arctic SDG framework would need attention to the relationship between national and international institutions vis-a-vis indigenous peoples and indigenous governance traditions [66]. In the SDG framework, indigenous rights are only mentioned in relation to food security and education. Also relevant are the relationships between Arctic and international institutions, especially given the dynamics in the relationship between the Arctic and the rest of the world [80,81]. Another issue relevant for monitoring progress towards the SDGs, or towards an Arctic equivalent, is the relationship between local and national politics. This has been repeatedly raised in scenario workshops and interviews we have conducted, often with a perception of the local level not being heard, e.g., [37]. It would be useful to include attention to this concern in an Arctic SDG framework, not least for ensuring the legitimacy of the SDG process as such.

*3.17. SDG 17. Strengthen the Means of Implementation and Revitalize the Global Partnership for Sustainable Development*

Goal 17 focusses on global international relations but also, in its technology component, includes attention to access to internet and to environmental technologies. Also covered are systemic issues

related to policy coherence and to data, monitoring and accountability. For an Arctic SDG framework, it would be equally relevant to highlight circumpolar partnerships. This could, for example, include attention to how well different countries fulfil the SDGs in their Arctic region and the progress of developing processes for making the SDG framework, or an Arctic SDG framework, relevant for local decision-making processes. We return to these issues in the Discussion.

## 4. Discussion and Conclusions

Sustainable development has been a topic in Arctic politics since the mid-1980s, including articulations of indigenous perspectives on sustainability and as a framing in the negotiations of circumpolar cooperation [25,82–85]. When the Arctic Council was established by the Ottawa Declaration in 1996, sustainable development became an official goal of the circumpolar cooperation [86], but, so far this mention has not been accompanied by any dedicated circumpolar assessment. In 2017, the Arctic Council's Ministerial Declaration reaffirmed the UN SDGs and the need for their realization by 2030, as well as the role of the Arctic Council in promoting sustainable development "through harmonizing its three core pillars in an integrated way: economic development, social development and environmental protection" [87]. However, it did not include any commitment to specific SDG follow-up.

Meanwhile, the research community has moved forward with attention to sustainability issues as well as to the need for new approaches for doing research in the Arctic [7,25]. Furthermore, and despite the apparent lack of political commitment to assessing progress towards the SDGs in the Arctic, Arctic scholars have recognized the need Arctic-relevant SDGs [14]. The exploratory analysis we have presented here about the match between the SDGs and relevant work on Arctic Social Indicators and other related research is a first step in this direction.

Our analysis shows that many of the SDGs and their sub-targets are highly relevant for assessing sustainable development in the Arctic but also that some well-recognized human development concerns are not well addressed in the SDGs. Table 1. Highlights some of the key issues that need additional attention in relation to each SDG.

Overall, an important missing aspect of the global SDGs in relation to the Arctic is the demographic challenges related to out-migration and urbanization, which both affect and are affected by many other issues. Another key issue that needs more attention than it receives in the SDG framework is indigenous rights, and more generally the right to control one's own future, or fate control as it is articulated in the ASI framework. A third concern is the need for better indicators (and data) on economic development that takes into account the specifics of Arctic economies, including the role of subsistence activities as well as characteristics of resource economics, such as their boom and bust-nature as well as their related financial flows [11].

A fourth issue relates to the importance of social capital highlighted in the literature of adaptation to climate change as a shorthand for the capacity of people to work together to solve problems [22,23]. While the notion is implicit in SDG 17 about global partnerships, social networks and well-functioning institutions and networks that can foster collaboration are needed at levels, from the local to the global. Furthermore, attention to the well-being of the international networks is essential at a time of increasing geopolitical tensions in the Arctic and elsewhere [82]. Increasing tensions over the use of landscapes and seascapes at a time of rapid environmental and social change places further needs on ensuring trust in how the institutions in society can manage conflicts of interests. We therefore suggest further discussion about how trust in institutions—at the local, national and international levels—could be captured by a framework for Arctic SDGs. This will be especially relevant in relation to management of resources that will be affected by the impacts of a warmer climate and by impacts of a necessary transition away from fossil fuels.

An overarching concern for future work on Arctic SDGs is data availability. A set of recommendations for data and statistics to enable and strengthen the unbiased measurement of ASI indicators was presented in the first ASI report, highlighting the critical need for improved data

availability, access to data, reporting of data and common data protocols across the Arctic, including a call for primary data collection to enable the full implementation of an ASI monitoring system [88].

**Table 1.** Summary of issues needing additional attention when constructing Arctic relevant targets and indicators for the global SDGs.

|    | Global SDG | Added Focus | Added Focus | Added Focus |
|----|-----------|-------------|-------------|-------------|
| 1  | No Poverty | Role of subsistence economy | Land and harvesting rights | Municipal economy |
| 2  | Zero Hunger | Role of country foods | Fishing, hunting and herding systems | Indigenous perspectives on food security |
| 3  | Good Health and Well-being | Demographic structure | | |
| 4  | Quality Education | Role of indigenous knowledge | Challenge of distances | |
| 5  | Gender Equality | Gendered out-migration | Men's changing roles | |
| 6  | Clean Water and Sanitation | Impacts of climate change | Impacts of industry | |
| 7  | Affordable Clean Energy | Role of financial incentives | | |
| 8  | Decent Work and Economic Growth | Volatility of resource economies | Economic leakage from the region | Role and contribution of subsistence and non-recognized stakeholders |
| 9  | Industry, Innovation and Infrastructure | Specifics of place | Impacts of climate change (e.g., permafrost thaw) | |
| 10 | Reduced inequalities | Regional policies | Financial flows | Indigenous rights |
| 11 | Sustainable Cities and Communities | Vulnerability to climate change | Urbanization | |
| 12 | Responsible Consumption and Production | Industrial waste | Local waste management | |
| 13 | Climate Action | | | |
| 14 | Life Below Water | Ice and changes in ice cover | | |
| 15 | Life on Land | Uniquely Arctic landscapes (e.g., tundra and high Arctic ecosystems) | Freshwater environments | Cumulative impacts |
| 16 | Peace, Justice and Strong Institutions | Indigenous peoples' rights | Indigenous governance traditions | |
| 17 | Partnership | Circumpolar cooperation | Trust in institutions | |

*Ways Forward*

The global SDG framework and its on-going implementation has put the spotlight on processes for monitoring and evaluating the success of political ambitions towards sustainable development. The work includes specific efforts to localize the SDGs [89], analyzing their interactions [90] and critical analyses of the actual roles of indicators, including how they can shape public debate, argumentation and shared understanding but also serve as tools for delaying action and legitimizing predetermined positions [33].

The exploratory assessment of the relevance of the SDGs for the Arctic provided in this article shows that local case studies and participatory processes that involve local and regional stakeholders can highlight aspects of sustainability that are not as prominent in the outcome of global political negotiations about SDGs. To further develop an indicator framework for assessing progress towards sustainable development in the Arctic requires further bottom–up process for mustering the knowledge and perspectives of indigenous peoples and a diversity of local communities. As pointed out in the

EU-PolarNet White paper, such processes are critical for making the SDGs relevant on the ground [14]. Even more important is that participatory processes related to the SDGs could provide spaces for learning and local capacity building for navigating rapid environmental and social change, similar to what has been shown with participatory scenario processes [36,91]. While this may be especially relevant in a region such as the Arctic, with a history of outsiders imposing their priorities and narratives about desirable futures on people living there, it is likely to be relevant across the world, as many decisions that will affect whether the SDGs will be achieved are made at the local and county levels of governance and by private actors (businesses as well as individuals).

We leave open the question of whether it would also be relevant with an Arctic SDG framework to guide decisions about the future of the region. However, some further work of Arctic SDGs is nevertheless relevant as support for subnational and local SDG processes within the region. Such work on Arctic SDGs would require interdisciplinary scientific effort in an endeavor that may need institutional support beyond what is possible to achieve in a time-limited research project. Potential candidates for providing such institutional context are the Arctic Council or a joint effort of the International Arctic Social Science Association (IASSA) and International Arctic Science Committee (IASC), which in turn would be able to bring in expertise from the Arctic Council working groups and their networks as well as from independent institutions in processes that focus on co-production of knowledge along the lines that are increasingly highlighted in the literature on Arctic sustainability research [25]. Such a project could also take responsibility for setting up a process that ensures the legitimacy of the efforts, relevant input, a sense of ownership, and the potential for exchange of knowledge and insights among a range of potential users of the Arctic SDGs, locally, nationally and internationally.

Arctic actors have taken a lead in developing regionally relevant assessment processes that speak to both global and regional challenges [92]. These include assessments of pollution [69,71,93], climate change [12,74], and human development [13,35]. Over time, and in combination with circumpolar scientific cooperation, they have fostered strong circumpolar scientific networks. These networks, together with a growing number of projects that focus on co-production of knowledge across scientific disciplines and with indigenous knowledge holders, place the Arctic in a good position to take the lead once again, this time with Agenda 2030 and the Sustainable Development Goals in focus.

**Author Contributions:** Conceptualization, A.E.N. and J.N.L.; writing—original draft preparation, A.E.N; writing—review and editing, J.N.L. All authors have read and agreed to the published version of the manuscript.

**Funding:** This research was funded by the NordForsk-funded Centre of Excellence "Resource Extraction and Sustainable Arctic Communities" under the programme Responsible Development of the Arctic: Opportunities and Challenges—Pathways to Action. J.N.L.'s work on the publication is also part of the Nunataryuk project, funded by the European Union's Horizon 2020 Research and Innovation Programme under grant agreement no. 773421.

**Acknowledgments:** We acknowledge the time and commitment provided by many people who have contributed insights in workshops and interviews. All authors have read and agreed to the published version of the manuscript.

**Conflicts of Interest:** The authors declare no conflict of interest. The funders had no role in the design of the study; in the collection, analyses, or interpretation of data; in the writing of the manuscript, or in the decision to publish the results.

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
