# Peer review of "Making Regional Sense of Global Sustainable Development Indicators for the Arctic"

_sustainability, doi:10.3390/su12031027_

Round 1
Reviewer 1 Report
Too much contractions (ASI, JNL, AEN...). Difficult to follow whilst reading. Perhaps sometimes repeating their complete names ...¿?
Results: is it possible to add some kind of table, abstracting the results and comparing chosen artic indicators with global ones... In order to be fairer. The text can contain the detail, the table could express the abstracted results.
Discussion:
Could you perhaps add some context? Is this explained processes happen in other parts of the world? comparison? It is important give some wider perspective to readers.
Perhaps too long paragraphs. Could you intend to 'break' them in more simple ideas? Too long paragraphs make readers to 'loose the line', the main argument.
Specific issues:
74. Two or three goals? Two? Three?
137-138. Complete reference.
Congrats! Great job, and quite needed, considering all this confussion about SDGs around the different regions, countries and performance measuring systems.
Author Response
Comment: Too much contractions (ASI, JNL, AEN...). Difficult to follow whilst reading. Perhaps sometimes repeating their complete names ...¿?
Response: We have spelled out JNL, AEN and AHDR but have left ASI and SDGs as these contractions occur repeatedly throughout the manuscript
Results:
Comment: is it possible to add some kind of table, abstracting the results and comparing chosen artic indicators with global ones... In order to be fairer. The text can contain the detail, the table could express the abstracted results.
Response: We have added a table in the Discussion and conclusions that summarizes the main additional issues that need attention in the Arctic
Discussion:
Comment: Could you perhaps add some context? Is this explained processes happen in other parts of the world? comparison? It is important give some wider perspective to readers.
Response: A paragraph has been added in the section “Ways forward. The text added in the Introduction in response to Reviewer 2 also provides additional context. We have not added text about similar processes in other parts of the world, as such material appear difficult to find in the published literature. There may be relevant initiatives, but it would require substantive work to provide a fair overview for comparison with our own Arctic study.
Comment: Perhaps too long paragraphs. Could you intend to 'break' them in more simple ideas? Too long paragraphs make readers to 'loose the line', the main argument.
Response: Text has been broken up into several paragraphs
Specific issues:
Two or three goals? Two? Three?
Response: Replaced with “more than one goal”
137-138. Complete reference.
Response: Contributing authors added to the reference
Reviewer 2 Report
The article Making Regional Sense of Global Sustainable Development Indicators for the Arctic” provides what the authors refer to as “an exploratory assessment” of what the global sustainable development goals and indicators look like from an Arctic perspective. The article deals with timely and important topics: well-being and development in the Arctic region, which is among the most rapidly changing regions in the world equally in terms of natural and societal change. My comments to develop the manuscript further can be found below:
In its current form, the manuscript is a heavily empirically oriented, loose comparison of existing research on Arctic sustainability and social change and the newest sustainable development goals. In practice, the authors argue that global SDGs are not able to grasp some specificities of life and development in the circumpolar north; hence, regionally sensitive indicators such as the ASI are needed and need to be further developed. By adopting this position, the authors also implicitly (although not wholly unreservedly) argue that a) the Arctic is 1) a uniform region and 2) that as a region it differs from other regions in the world. While based on this position the relevance of global indicators can be questioned, what also could be questioned is that whether regional approaches are the way forward either. Can the specificities of development in the internally heterogenous Arctic indeed be grasped by the same indicators everywhere in the region or could it be that e.g. the cities in the region could be compared by the indicators to measure the sustainability of cities worldwide or that the indicators required to understand life and development in the rural Arctic might bear relevance to the requirements of rural sustainability outside the Arctic region? While it remains up to the authors to justify their stance towards the issue, I warmly urge them to deal with the questions above.
The manuscript’s theoretical underpinnings are discussed only superficially; I feel the paper is lacking proper (even brief!) theoretical discussion about sustainability (which does not equal sustainable development, although the authors appear the use the term interchangeably). This lack of theoretical and conceptual reflexivity is also demonstrated by the authors’ rather unreservedly positive stance towards developing and using indicators in the first place and the lack of references to discussions that question the whole rather managerialistic practice of measuring. In a similar manner, the complex concept of resilience is not even briefly defined, only illustrated by some findings from the Arctic Resilience Report. The paper would greatly benefit from engaging even in some critical theoretical and conceptual discussion – the seeds for this are already present in the paper in its current form.
Despite the criticism presented above, the manuscript deals with societally and academically relevant concerns and can serve as a platform for future discussion. Despite its superficial engagement with concepts, theories or methods that would normally be required from a research article, its results and conclusions do a good job in positioning the existing research on Arctic-specific indicators and sustainable development within the broader discussion on SDGs on vice versa. As such, the article serves as a commentary to ongoing discussions and developments.
Author Response
Comment: In its current form, the manuscript is a heavily empirically oriented, loose comparison of existing research on Arctic sustainability and social change and the newest sustainable development goals. In practice, the authors argue that global SDGs are not able to grasp some specificities of life and development in the circumpolar north; hence, regionally sensitive indicators such as the ASI are needed and need to be further developed. By adopting this position, the authors also implicitly (although not wholly unreservedly) argue that a) the Arctic is 1) a uniform region and 2) that as a region it differs from other regions in the world. While based on this position the relevance of global indicators can be questioned, what also could be questioned is that whether regional approaches are the way forward either. Can the specificities of development in the internally heterogenous Arctic indeed be grasped by the same indicators everywhere in the region or could it be that e.g. the cities in the region could be compared by the indicators to measure the sustainability of cities worldwide or that the indicators required to understand life and development in the rural Arctic might bear relevance to the requirements of rural sustainability outside the Arctic region? While it remains up to the authors to justify their stance towards the issue, I warmly urge them to deal with the questions above.
Response: The reviewer’s comment is very valid and relevant. We have therefore clarified our position by adding text in the Introduction and revising the Discussion.
Comment: The manuscript’s theoretical underpinnings are discussed only superficially; I feel the paper is lacking proper (even brief!) theoretical discussion about sustainability (which does not equal sustainable development, although the authors appear the use the term interchangeably). This lack of theoretical and conceptual reflexivity is also demonstrated by the authors’ rather unreservedly positive stance towards developing and using indicators in the first place and the lack of references to discussions that question the whole rather managerialistic practice of measuring. In a similar manner, the complex concept of resilience is not even briefly defined, only illustrated by some findings from the Arctic Resilience Report. The paper would greatly benefit from engaging even in some critical theoretical and conceptual discussion – the seeds for this are already present in the paper in its current form.
Response: Substantial new text, including the section’ Theoretical context’ has been added to the Introduction to address these comments.